# ADAPTIVE ONLINE PLANNING FOR CONTINUAL LIFE-LONG LEARNING

## ABSTRACT

We study learning control in an online lifelong learning scenario, where mistakes can compound catastrophically into the future and the underlying dynamics of the environment may change. Traditional model-free policy learning methods have achieved successes in difficult tasks due to their broad flexibility, and capably condense broad experiences into compact networks, but struggle in this setting, as they can activate failure modes early in their lifetimes which are difficult to recover from and face performance degradation as dynamics change. On the other hand, model-based planning methods learn and adapt quickly, but require prohibitive levels of computational resources. Under constrained computation limits, the agent must allocate its resources wisely, which requires the agent to understand both its own performance and the current state of the environment: knowing that its mastery over control in the current dynamics is poor, the agent should dedicate more time to planning. We present a new algorithm, Adaptive Online Planning (AOP), that achieves strong performance in this setting by combining model-based planning with model-free learning. By measuring the performance of the planner and the uncertainty of the model-free components, AOP is able to call upon more extensive planning only when necessary, leading to reduced computation times. We show that AOP gracefully deals with novel situations, adapting behaviors and policies effectively in the face of unpredictable changes in the world – challenges that a continual learning agent naturally faces over an extended lifetime – even when traditional reinforcement learning methods fail.

## 1 INTRODUCTION

We consider agents in a human-like setting, where agents must simultaneously act and learn in the world continuously with limited computational resources. All decisions are made online; there are no discrete episodes. Furthermore, the world is vast – too large to feasibly explore exhaustively – and changes over the course of the agent's lifetime, like how a robot's actuators might deteriorate with continued use. There are no resets to wipe away past errors. Mistakes are costly, as they compound downstream. To perform well at reasonable computational costs, the agent must utilize its past experience alongside new information about the world to make careful, yet performant, decisions.

Non-stationary worlds require algorithms that are fundamentally robust to changes in dynamics. Factors that would lead to a change in the environment may either be too difficult or principally undesirable to model: for example, humans might interact with the robot in unpredictable ways, or furniture in a robot's environment could be rearranged. Therefore, we assume that the world can change unpredictably in ways that cannot be learned, and focus on developing algorithms that instead handle these changes gracefully, without using extensive computation.

Model-based trajectory optimization via planning is useful for quickly learning control, but is computationally demanding and can lead to bias due to the finite planning horizon. Model-free reinforcement learning is sample inefficient, but capable of cheaply accessing past experience without sacrifices to asymptotic performance. Consequently, we would like to distill expensive experience from an intelligent planner into neural networks to reduce computation for future decision making.

Deciding when to use the planner vs a learned policy presents a difficult challenge, as it is hard to evaluate the improvement the planner would give without actually using the planner. We tackle this as a problem of uncertainty. When uncertain about a course of action, humans use an elongated

model-based search to evaluate long-term trajectories, but fall back on habitual behaviors learned with model-free paradigms when they are certain of what to do (Dayan & Berridge, 2014; Daw et al., 2005; Banks & Hope, 2014; Kahneman, 2003). By measuring this uncertainty, we can make informed decisions about when to use model-based planning vs a model-free policy.

Our approach combines model-based planning with model-free policy learning, along with an adaptive computation mechanism, to tackle this setting. Like a robot that is well-calibrated when first coming out of a factory, we give the agent access to a ground truth dynamics model that lacks information about future changes to the dynamics, like different settings in which the robot may be deployed. This allows us to make progress on finite computation continual learning without worrying about model learning. The dynamics model is updated immediately at world changes. However, as we show empirically, knowing the dynamics alone falls far short of success at this task.

We present a new algorithm, Adaptive Online Planning (AOP), that links Model Predictive Path Integral control (MPPI) (Williams et al., 2015), a model-based planner, with Twin Delayed DDPG (TD3) (Fujimoto et al., 2018b), a model-free policy learning method. We combine the model-based planning method of iteratively updating a planned trajectory with the model-free method of updating the network weights to develop a unified update rule formulation that is amenable to reducing computation when combined with a switching mechanism. We inform this mechanism with the uncertainty given by an ensemble of value functions. Access to the ground truth model is not sufficient by itself, as we show that PPO (Schulman et al., 2017) and TD3 perform poorly, even with the ground truth model. We demonstrate empirically that AOP is capable of integrating the two methodologies to reduce computation while achieving and maintaining strong performance in non-stationary worlds, outperforming other model-based planning methods and avoiding the empirical performance degradation of policy learning methods in changing world scenarios.

Our contributions include the proposal of a new algorithm combining model-based planning and model-free learning, the introduction of evaluation environments that target the challenges of life-long reinforcement learning in which traditional methods struggle, and experiments showing the usefulness of utilizing both model-based and model-free methods in this setting. Code to run all algorithms and experiments is available at `www.github.com/` (link redacted for anonymity).

## 2 BACKGROUND

We consider the world as an infinite-horizon Markov Decision Process (MDP), defined by the tuple $\mathcal{M} = \{\mathcal{S}, \mathcal{A}, \mathcal{R}, \mathcal{T}, \gamma\}$, where $\mathcal{S}$ is the state space, $\mathcal{A}$ is the action space, $\mathcal{R} : \mathcal{S} \times \mathcal{A} \to \mathbb{R}$ is the reward function, $\mathcal{T} : \mathcal{S} \times \mathcal{A} \times \mathcal{S} \to \mathbb{R}$ are the transition probabilities, and $\gamma \in [0, 1)$ is the discount factor. The world can change over time: the transitions and rewards $(\mathcal{T}, \mathcal{R})$ may change to some new $(\mathcal{T}', \mathcal{R}')$. Unlike traditional reinforcement learning, the agent's state is not reset at these world changes. The agent can generate rollouts using the current $(\mathcal{T}, \mathcal{R})$ starting from its current state, but not for future $(\mathcal{T}', \mathcal{R}')$. The agent's goal is to execute a future sequence of actions, summarized as a policy $\pi(a_t|s_t)$, that maximizes the expected future return $R(t) = \mathbb{E}_{\tau \sim \pi}[\sum_{k=0}^{\infty} \gamma^k r(s_t, a_t)]$.

### 2.1 CONTINUAL ONLINE LEARNING

In our work, we consider online learning in the same style as Lowrey et al. (2018), where both acting and learning must occur on a per-timestep basis, and there are no episodes that reset the state. At each timestep, the agent must execute its training procedure, and is then forced to immediately output an action. We also desire agents that are equipped, like humans, to handle different tasks in various environments. Continual learning is difficult, as agents must use their experience to learn to perform well in new tasks (forward transfer) while preserving the ability to perform well in old tasks (backward transfer). In addition to the these difficulties, there is also the challenge of avoiding failure sink states that prevent future learning progress. We augment this task with a world where the dynamics continually change, creating a difficult setting for agent learning.

### 2.2 MODEL-BASED PLANNING

Online model-based planning evaluates future sequences of actions using a model, develops a projected future trajectory over some time horizon, and then executes the first action of that trajectory,

before repeating. We specifically focus on Model Predictive Control (MPC), which iteratively applies Gaussian noise to the prior predicted trajectory, evaluates them using the dynamics model, and combines them with an update rule. When the update rule is a softmax weighting, this procedure is called Model Predictive Path Integral (MPPI) control (Williams et al., 2015). Due to the nature of this iterative and extended update, this procedure is computationally expensive.

### 2.3 Model-Free Policy Optimization

Model-free algorithms encode the agent's past experiences in a function dependent only on the current state, often in the form of a value function critic and/or policy actor. As a result, such algorithms can have difficulty learning long-term dependencies and struggle early on in training; temporally extended exploration is difficult. In exchange, they attain high asymptotic performance, having shown successes in a variety of tasks for the traditional offline setting (Mnih et al., 2013; Schulman et al., 2017). As a consequence of their compact nature, once learned, these algorithms tend to generate cyclic and regular behaviors, whereas model-based planners have no such guarantees.

We run online versions of TD3 (Fujimoto et al., 2018b) and PPO (Schulman et al., 2017) as baselines to AOP. While there is no natural way to give a policy access to the ground truth model, we allow the policies to train on future trajectories generated via the ground truth model, in similar fashion to algorithms that learn a model for this purpose (Kurutach et al., 2018; Buckman et al., 2018), in order to help facilitate fair comparisons to model-based planners.

### 2.4 Update Rule Perspective on Planning vs Policy Optimization

From a high-level perspective, the model-based planning and model-free policy optimization procedures are very similar (see Appendix B for a side-by-side comparison). Where the planner generates noisy rollouts to synthesize a new trajectory, the model-free algorithm applies noise to the policy to generate data for learning. After an update step, either an action from the planned trajectory or one call of the policy is executed. These procedures are only distinct in their respective update rules.

The primary contribution of AOP is unifying both update rules to compensate for their individual weaknesses. AOP distills the learned experience from the planner into the off-policy learning method of TD3 and a value function, so that planning and acting can be done cheaper in the future.

## 3 Adaptive Online Planning

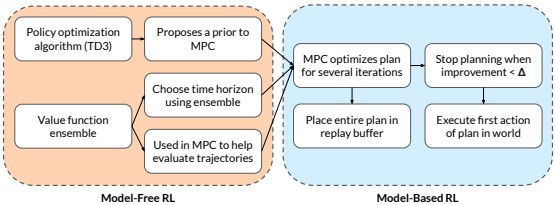

Figure 1: Schematic view of Adaptive Online Planning (AOP).

AOP combines model-based planning with model-free learning using an adaptive switching mechanism, summarized above in Fig. 1, and written with more detail in Appendix B in Alg. 3.

### 3.1 Model-Based Planning with Terminal Value Approximation

For the model-based component, AOP uses MPPI, as described in Section 2.2, with a terminal value function $\hat{V}$, where trajectories are evaluated in the form of Eq. 1. This process is repeated for several iterations to improve the plan, and then the first action is executed in the environment.

$$R(\tau) = \sum_{k=0}^{H-1} \gamma^k r(s_k, a_k) + \gamma^H \hat{V}(s) \tag{1}$$

$\hat{V}$ is generated by an ensemble of $n$ value functions (see Eq. 2), as proposed in POLO (Lowrey et al., 2018) for MPC. The value ensemble improves the exploration ability of the optimization procedure (Osband et al., 2016; 2018). The log-sum-exp function serves as a softmax, enabling exploration while preserving a stable target for learning. The $\log n$ term normalizes the estimate to lie between the mean and the max of the ensemble, determined by the temperature hyperparameter $\kappa$, which ensures that the approximation is still semantically meaningful for estimating the value of the state.

$$\hat{V}(s) = \frac{1}{\kappa} \log \sum_{i=1}^{n} e^{\kappa V_i(s) - \log n} \tag{2}$$

### 3.2 EARLY PLANNING TERMINATION

Past model-based planning procedures (Chua et al., 2018; Wang & Ba, 2019) run a fixed number of iterations of MPC per timestep before executing an action in the environment. However, this is often wasteful. Within a particular timestep, later planning iterations often improve the planned trajectory less than earlier iterations, and may not even improve the trajectory at all. We propose to decide on the number of planning iterations on a per-timestep basis. After generating a new trajectory $\tau_{k+1}$ from the $k$-th iteration of planning, we measure the improvement $\Delta(\tau_{k+1}|\tau_k)$ against the trajectory $\tau_k$ of the previous iteration (Eq. 3). When this improvement decreases below a threshold $\Delta_{thres}$, we terminate planning for the current timestep with probability $1 - \epsilon_{plan}$. Using a stochastic termination rule allows for robustness against local minima where more extensive planning may be required, but not evident from early planning iterations, in order to escape.

$$\Delta(\tau_{k+1}|\tau_k) = \frac{R(\tau_{k+1}) - R(\tau_k)}{|R(\tau_k)|} \tag{3}$$

### 3.3 ADAPTIVE PLANNING HORIZON

Since planning over a long time horizon is expensive, it would also be desirable to plan over a shorter time horizon when the planner is confident in achieving long-term success with only short-term planning. We represent the planner's uncertainty with the value ensemble from Section 3.1, as the mean and standard deviation of the ensemble represent the epistemic uncertainty of the value of the state (Osband et al., 2018). When deciding to use a reduced time horizon, we require that the standard deviation $\sigma$ of the value ensemble on the current state be lower than some threshold $\sigma_{thres}$.

A problem with only considering the standard deviation is that the metric only considers uncertainty with respect to the past – it does not immediately measure uncertainty in a changing dynamics setting, which is only observed when considering experiences in the future. Therefore, we fine-tune the horizon length using the Bellman error (Eq. 4). The time horizon is given by the largest $H \leq H_{full}$ such that $\epsilon(H|\tau_k) > \epsilon_{thres}$. When $\sigma > \sigma_{thres}$, the full time horizon is always used, regardless of the Bellman error. This is to say that, if the value function can accurately approximate the latter part of the horizon, we can use the value function instead. While choices for these hyperparameters are somewhat arbitrary, we show in Appendix C.1.1 that AOP is not particularly sensitive to them.

$$\epsilon(H|\tau_{s_t,\ldots,s_{t+H_{full}}}) = (R(\tau_{s_{t+H},\ldots s_{t+H_{full}}}) - \frac{1}{n} \sum_{i=1}^{n} V_i(s_{t+H}))^2 \tag{4}$$

### 3.4 OFF-POLICY MODEL-FREE PRIOR

We use TD3 as a prior to the planning procedure, with the policy learning off of the data generated by the planner during planning, which allows the agent to recall past experience quickly. Similarly to past work (Rajeswaran et al., 2017a; Zhu et al., 2018), we found that imitation learning can cap the asymptotic performance of the learned policy. As a baseline, we also run behavior cloning (BC) as a prior, and refer to the resulting algorithms as AOP-TD3 and AOP-BC, respectively.

We note that MPC and policy optimization are both special cases of AOP. MPC is equivalent to AOP with a constant setting for the time horizon that always uses full planning iterations (i.e. a threshold

of 0). Policy optimization is equivalent to AOP with one planning iteration, since the first plan is a noisy version of the policy, acting as the data collection procedure in standard policy learning.

## 4  EMPIRICAL EVALUATIONS

We investigate several questions empirically:

1. What are the challenges in the continual lifelong learning setting? When developing further algorithms in this setting, what should we focus on?

2. How does AOP perform when acting from well-known states, novel states, and in changing worlds? How do traditional on-policy and off-policy methods fare in these situations?

3. Are the variance and the Bellman error of the value ensemble suitable metrics for determining the planning computational budget?

### 4.1  LIFELONG LEARNING ENVIRONMENTS

We propose three environments to evaluate our proposed algorithm in the continual lifelong learning setting: Hopper, Ant, and Maze, and release them for others to use. While these environments are not overly complex control environments, they crisply highlight the difficulties of continual lifelong learning in MDPs. Pictures of these environments are included in Appendix D.

**Hopper:** First, we consider the OpenAI Gym environment Hopper (Brockman et al., 2016). The agent is rewarded based on how closely it matches an unobserved target velocity. Every 4000 timesteps, this target velocity changes. The environment is tough, as it can be difficult to get up if the agent falls down in a strange position, and momentum from past actions affect the state greatly, which makes it easy for the agent to fall over. We consider three versions of the Hopper environment: (1) a standard Hopper with an unchanging target velocity, (2) a novel states Hopper with the target velocity in the observation (thus new target velocities correspond to the agent seeing a new state), and (3) a changing worlds Hopper, where the target velocity is not in the observation.

**Ant**: We also consider the Ant from Gym. The agent seeks to maximize its forward velocity, but a joint at random is disabled every 2000 timesteps. Once flipped over, getting back up is extremely difficult, which makes this environment harshly unforgiving. We consider two versions: (1) a standard Ant with no disabled joints, and (2) a changing worlds Ant with one changing disabled joint.

**Maze:** Like in POLO, we test in a 2D point mass maze, where agent seeks to reach a goal. The observation is $(x_{point}, y_{point}, x_{goal}, y_{goal})$. Every 500 timesteps, the walls of the maze change, and the goal swaps locations. The difficulty lies in adapting quickly to new mazes while avoiding the negative transfer of old experience. We consider two versions: (1) a novel states Maze, where the walls of the maze remain constant, but new goals are introduced after 20 goal changes in the original positions, and (2) a changing worlds Maze, which is as described above. We also test both versions in a dense reward and a sparse reward setting, where the reward is either the negative L2 distance or a boolean value, respectively. In the sparse reward Maze, exploration can be particularly challenging.

### 4.2  BASELINES AND ABLATIONS

We run AOP-BC, POLO, MPC, TD3, and PPO as baselines against AOP-TD3; they can be seen as ablations/special cases of our proposed algorithm (see Section 3.4). We consider two versions of MPC, with 8 and 3 planning iterations, henceforth referred to as MPC-8 and MPC-3, respectively.

Table 1: Timesteps rolled out by planner (planning levels) as a fraction of MPC-8 for model-based planning algorithms. Shown are the average over all environments and the range (min-max) across the environments for 5 seeds. For more detailed graphs, see Fig. A.1 in Appendix A.

| AOP-TD3 | AOP-BC | POLO | MPC-8 | MPC-3 |
|---|---|---|---|---|
| **11.39% (1.40% - 16.62%)** | **11.40% (2.86% - 15.17%)** | 37.50% | 100% | 37.50% |

Table 2: Average lifetime rewards. S, NS, and CW, denote the standard, novel states, and changing worlds settings; (D) and (S) denote dense/sparse reward mazes. Shown is the average for 5 seeds with two standard deviations. Best results are bolded. See Appendix A for full learning curves.

| Environment | AOP-TD3 | AOP-BC | POLO | TD3 | PPO | MPC-8 | MPC-3 |
|---|---|---|---|---|---|---|---|
| S Hopper | $0.12 \pm 0.16$ | $\textbf{0.33} \pm \textbf{0.22}$ | **0.51** | 0.23 | -14.41 | 0.36 | 0.19 |
| NS Hopper | $\textbf{0.41} \pm \textbf{0.18}$ | $\textbf{0.53} \pm \textbf{0.18}$ | **0.59** | **0.40** | -14.22 | -0.28 | -0.49 |
| CW Hopper | $\textbf{0.48} \pm \textbf{0.24}$ | $\textbf{0.45} \pm \textbf{0.12}$ | **0.57** | -2.42 | -13.14 | -0.30 | -0.48 |
| S Ant | $3.02 \pm 0.13$ | $\textbf{3.38} \pm \textbf{0.27}$ | **3.40** | 2.19 | n/a | **3.52** | **3.40** |
| CW Ant | $2.76 \pm 0.47$ | $\textbf{3.11} \pm \textbf{0.41}$ | 2.90 | 2.05 | n/a | **3.32** | **3.14** |
| NS Maze (D) | $\textbf{-0.21} \pm \textbf{0.08}$ | $\textbf{-0.25} \pm \textbf{0.02}$ | **-0.25** | -1.81 | -2.14 | **-0.19** | **-0.25** |
| CW Maze (D) | $-0.29 \pm 0.07$ | $-0.34 \pm 0.03$ | -0.30 | -1.17 | -2.10 | **-0.19** | -0.30 |
| NS Maze (S) | $\textbf{0.85} \pm \textbf{0.07}$ | $0.70 \pm 0.06$ | 0.62 | -0.68 | -0.88 | 0.69 | 0.61 |
| CW Maze (S) | $\textbf{0.69} \pm \textbf{0.20}$ | $\textbf{0.56} \pm \textbf{0.04}$ | 0.57 | -0.66 | -0.74 | **0.58** | **0.52** |

(a) Changing worlds Hopper     (b) Changing worlds Ant     (c) Changing worlds Maze (D)

Figure 2: Reward curves for changing worlds lifelong learning tasks. Rewards are for a single timestep, not over an episode. Note that some worlds may be more difficult than others, and yield a naturally lower reward. The results are averaged over 5 seeds; the shaded area depicts one standard deviation above and below the mean. Curves are smoothed and the rewards are clipped to -3 for visual clarity. See Fig. A.2 in Appendix A for corresponding plots for all environments.

### 4.3 CHALLENGES IN CONTINUAL LIFELONG LEARNING SETTING

Planner usage is shown in Table 1 and rewards are in Table 2. AOP uses only $1 - 17\%$ of the number of timesteps as MPC-8, but achieves generally comparable or stronger performance in most environments. More detailed graphs can be found in Appendix A.

**Reset-Free Setting:** Even with model access, these environments are challenging for the algorithms to learn. In the standard offline reinforcement learning setting, long-term action dependencies are learned from past experience over time, and this experience can be utilized when the agent resets to the initial state. However, in the online setting, these dependencies must be learned on the fly, and if the agent falls, it must return to the prior state in order to use that information. In particular, for the Ant environment, such falling is catastrophic, as it takes a complex action sequence to return to standing. POLO-style optimistic exploration can thus be a disadvantage, encouraging the Ant to take on new and unstable behaviors. In spite of this, AOP, with about $39\%$ of the planning of POLO, achieves comparable performance to POLO; AOP-BC achieves very strong performance, in general.

**Vast Worlds:** The performance gain of MPC-8 over MPC-3 shows that achieving strong performance is difficult with constrained computation. In the sparse mazes, MPC is significantly outperformed by AOP-TD3, and the model-free algorithms struggle to make any progress at all, showing their lackluster exploration. Even POLO – the exploration mechanism of AOP – faces weaker performance, indicating that AOP-TD3 has not only correctly identified when planning is important, but is able to effectively leverage additional computation to increase its performance whilst still using less overall computation. The additional performance in the novel states Maze (S) over MPC-8 also shows AOP's ability to consolidate experience to improve performance in mazes it has seen before. Furthermore, in the changing worlds Maze (S), the performance of AOP improves over time (Fig. A.2), indicating that AOP has learned value and policy functions for effective forward transfer.

**Policy Degradation:** TD3's performance significantly degrades in the changing worlds settings, as does PPO's (see Fig. 2). PPO, an on-policy method, struggles in general. In the novel states Hopper, the variant where the policy is capable of directly seeing the target velocity, TD3 performs very well, even learning to outperform MPC. However, without the help of the observation, in the changing worlds, TD3's performance quickly suffers after world changes. The model-based planning methods do not suffer this degradation, and AOP is able to maintain its performance and computational savings, even through many world changes, despite its reliance on model-free components.

## 4.4 BEHAVIOR OF POLICIES IN CONTINUAL LIFELONG LEARNING SETTING

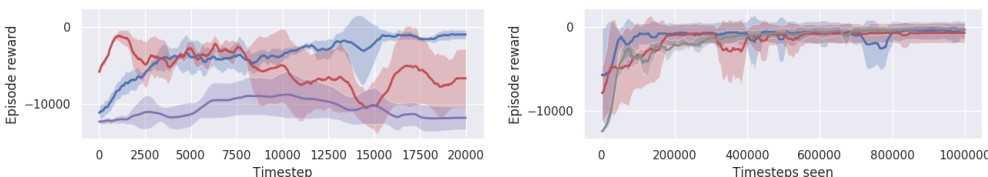

Figure 3: Policy episode performance for changing worlds Hopper. Left: performance of policy only (no planner) throughout training. Right: performance on initial starting state and target velocity after additional TD3 training (blue: AOP-TD3, red: TD3, gray: from scratch). 5 seeds are shown.

In Fig. 3 (left), we plot the episodic reward of the policy running from the agent's current state for each timestep (for the current target velocity). Note that since the AOP policy is learned from off-policy data (the planner), it suffers from divergence issues and should be weaker than TD3 on its own (Fujimoto et al., 2018a). Matching the result in Fig. 2 (a), the TD3 policy degrades in performance over time, but the AOP policy does not. This suggests that the policy degradation effect might stem from exploration, rather than from an issue with the optimization algorithm.

We also show in Fig. 3 (right) tuning the policy learned by AOP after seeing every target velocity once (blue) vs. by TD3 (red) vs. training a new policy (gray), learning from running the standard episodic TD3 algorithm on the first target velocity. The AOP policy learns much faster, showing that AOP is capable of quick backward transfer and adapting quickly to a different situation.

## 4.5 BEHAVIOR OF AOP IN WELL-KNOWN STATES/NOVEL STATES/CHANGING WORLDS

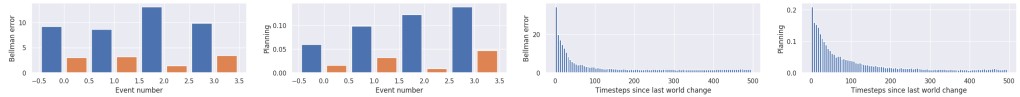

Figure 4: Graphs for Maze (D). From left to right: (1-NS): average Bellman error of the the first time a goal is presented (blue) vs the last time it is presented (orange). (2-NS) average number of planning steps. Events denote the 4 times the set of goals switch during the agent's lifetime. (3-CW): average Bellman error by the time since the last world change. (4-CW): average number of planning steps. Both quickly decrease as the agent becomes more familiar with the world.

Fig. 4 shows AOP behavior in Maze (D). When encountering novel states, Bellman error is high, but as time progresses, when confronted with the same states again, Bellman error becomes low. The number of planning timesteps matches this: AOP correctly identifies a need to plan early on, but greatly saves computation later, when it is immediately able to know the correct action with almost no planning. The same effect occurs when measuring the time since the world changed for the changing worlds. At the beginning of a new world, the amount of planning is high, before quickly declining to nearly zero, almost running with the speed of a policy: $\approx 100\times$ faster than MPC-8.

We plot the standard deviation and Bellman error over time of AOP for the changing worlds Hopper in Fig. 5. After each world change, the Bellman error spikes, and then decreases as time goes on. These trends are reflected in the time horizon (bottom center), which decreases as the agent trains in each world, and indicate that the standard deviation and Bellman error are suitable metrics for determining planning levels. The same effect also occurs for the number of planning iterations.

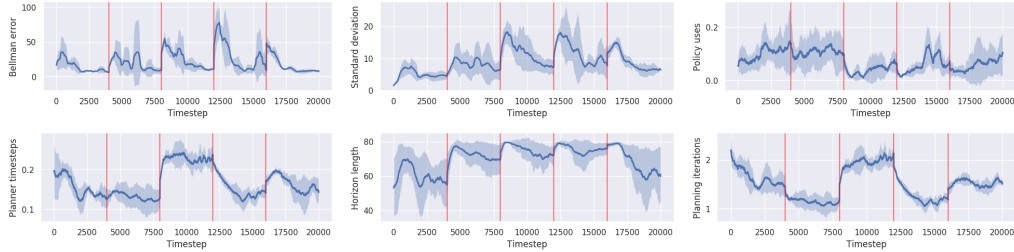

Figure 5: Graphs for AOP in changing worlds Hopper. Red lines denote world changes. Policy uses is the percent of the time that the policy was used instead of the initial plan (see line 3 of Alg. 3).

In Fig. 6, we show qualitatively how TD3, MPC, and AOP handle world changes in the changing worlds Hopper setting. An unobserved change in the target velocity from 2.5 to 1 is encountered during the agent's lifetime. The purely model-free TD3 produced cyclic and regular behavior, but adapted slowly to the target speed, still moving at 2.5. MPC quickly adapted to the new target speed, but moved in an irregular fashion. AOP was both able to rapidly adapt and do so in a regular manner.

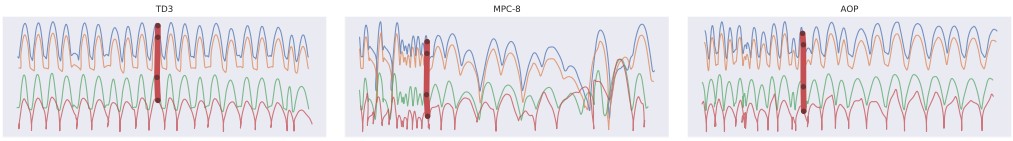

Figure 6: Example trajectory traces of Cartesian hopper positions for different algorithms. An unobserved change in target speed is encountered at the timestep marked by red outline.

## 5    RELATED WORK AND FUTURE DIRECTIONS

**Continual online learning:** Much of past lifelong learning work (Goodfellow et al., 2013; Parisi et al., 2018) has focused on catastrophic forgetting, which AOP is resilient to, but was not a primary focus of our work. Kearns & Singh (2002) considers MDPs with various reward functions, using them to make conscious decisions about exploration or exploitation, similar to our framework. Finn et al. (2019) uses meta-learning to perform continual online learning, but the tasks are considered in episodes. Nagabandi et al. (2018) learn multiple models to represent different tasks, which contrasts with our single unified policy/set of value functions.

**Augmented planners:** Algorithms that combine planning with learning have previously been studied in both discrete and continuous domains (Chua et al., 2018; Anthony et al., 2017). Recent work (Guez et al., 2018) generalizes the MCTS algorithm and proposes to learn the algorithm instead; further asking the algorithm to set computation levels could be effective in our setting as well. Levine & Koltun (2013); Mordatch et al. (2015) propose to use priors that make trajectory planner stay close to policy outputs, which is problematic in changing worlds when the policy is not accurate.

**Learned dynamics:** Many previous works (Azizzadenesheli et al., 2018; Nagabandi et al., 2017) have learned dynamics models and then performed MPC optimization on them. Kurutach et al. (2018); Clavera et al. (2018) utilized model ensembles to reduce performance degradation due to model overfitting, and (Janner et al., 2019) investigates model uncertainty for policy learning. Integrating AOP with a learned uncertainty-aware dynamics model would be interesting future work.

## 6    CONCLUSION

We proposed AOP, which incorporates model-based planning with model-free learning, and introduced environments for evaluating algorithms in the continual lifelong learning setting. We empirically analyzed the performance of and signals from the model-free components, and showed experimentally that AOP was able to successfully reduce computation while achieving high performance in difficult tasks, often competitive with a much more powerful MPC procedure.

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

# A    DETAILED EXPERIMENTAL GRAPHS

Additional graphs are provided for the fraction of planning timesteps vs MPC-8 and reward curves for all environments in Figures A.1 and A.2, respectively.

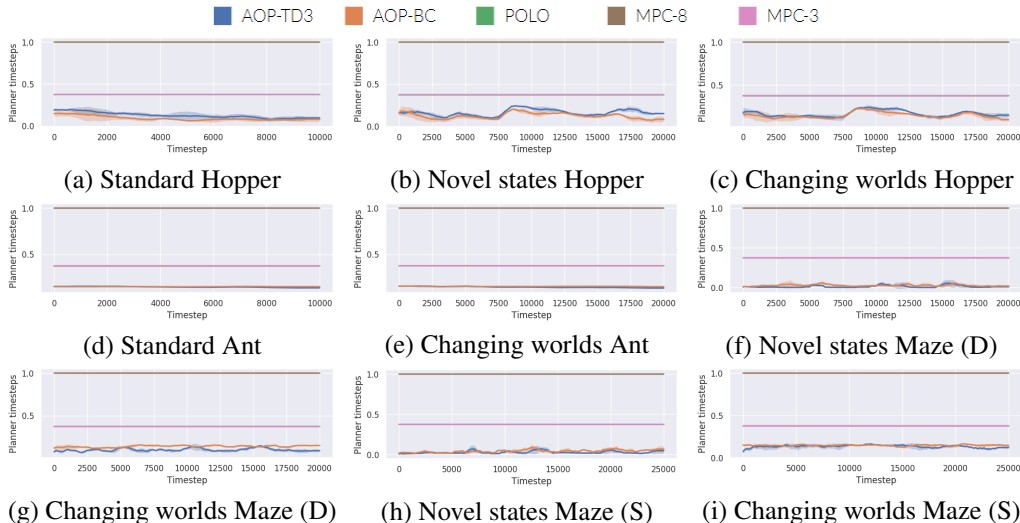

Figure A.1: Number of timesteps rolled out by planner per timestep as a percentage of MPC-8.

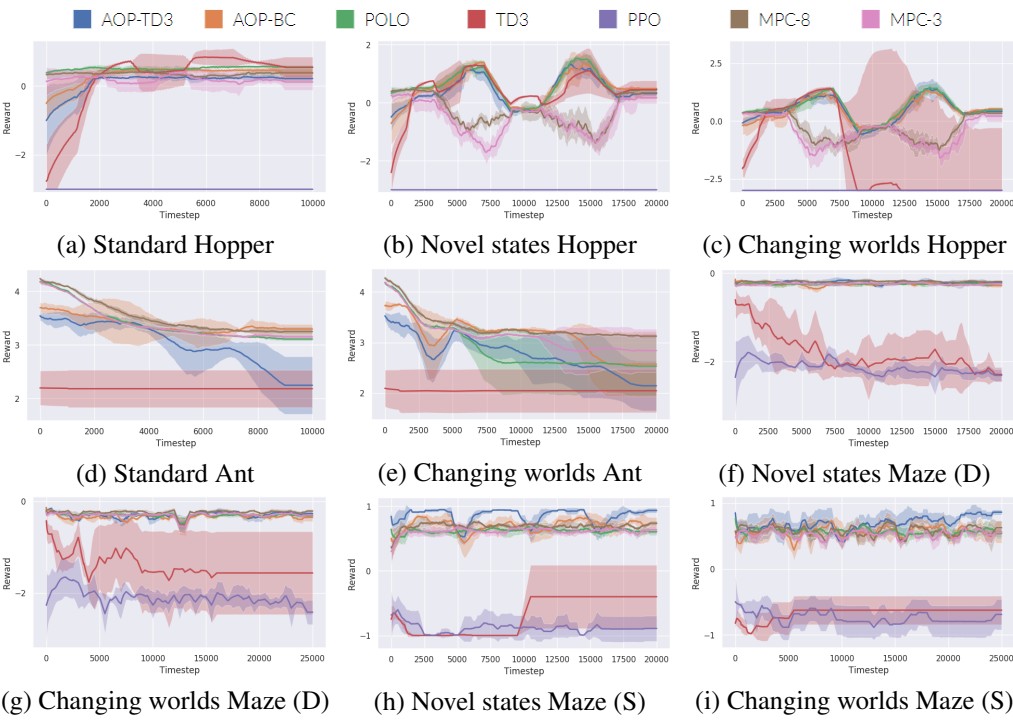

Figure A.2: Reward curves for lifelong learning tasks.

## B    Pseudocode for Algorithms

### B.1    Model-Based Planning vs Policy Optimization Pseudocode

| **Algorithm 1:** Model-Based Planning | **Algorithm 2:** Policy Optimization |
|---|---|
| 1  Initialize action trajectory $\tau_{plan}$ | 1  Initialize policy $\pi_\phi$ |
| 2  **while** *alive* **do** | 2  **while** *alive* **do** |
| 3   Generate $n$ rollouts based on $\tau_{plan}$ | 3   Generate $n$ rollouts based on $\pi_\phi$ |
| 4   Use rollouts to update $\tau_{plan}$ | 4   Use rollouts to update $\pi_\phi$ |
| 5   Execute first action of $\tau_{plan}$ | 5   Execute an action from $\pi_\phi$ |
| 6  **end** | 6  **end** |

### B.2    Adaptive Online Planning Pseudocode

**Algorithm 3:** Adaptive Online Planning

1  Initialize value ensemble $\{V_i\}_{i=1}^n$, policy $\pi$, replay buffers $\mathcal{D}_V, \mathcal{D}_\pi$
2  **while** *alive* **do**
3   Set $\tau_{plan}$ to $\arg\max_{\tau_\pi, \tau_{plan}} R(\tau)$, generating $\tau_\pi$ from the policy $\pi_\theta$
4   Select time horizon $H_t$ for planning, described in Section 3.3
5   **for** $k \leftarrow 1$ **to** $max\_iters$ **do**
6    Run MPC planning to generate $\tau_{plan}$, described in Section 3.1; add trajectories to $\mathcal{D}_\pi$
7    If $\Delta(\tau_{k+1}|\tau_k) < \Delta_{thres}$, stop planning with probability $1 - \epsilon_{plan}$
8   **end**
9   If it's time to update: update value ensemble $\{V_i\}_{i=1}^n$ and policy $\pi_\theta$ with $\mathcal{D}_V, \mathcal{D}_\pi$, resp.
10  Step once in the environment with first action of $\tau_{plan}$; add $(s_t, a_t, s_t', r_t)$ to $\mathcal{D}_V$
11 **end**

## C    Hyperparameter Details

All of our implementations and hyperparameters are available at www.github.com/ (link redacted for anonymity).

### C.1    Adaptive Online Planning Hyperparameters

For AOP, we set $\sigma_{thres} = 8$, $\epsilon_{thres} = 25$ and $\epsilon_{plan} = 0.2$. We did not tune these hyperparameters much, and similarly do not believe that the algorithm is overly sensitive to the thresholds in dense reward environments (see Appendix C.1.1). However, in the sparse Mazes, we set $\sigma_{thres} = \epsilon_{thres} = 0$, in order to avoid early termination of exploration (we do not change the hyperparameters determining the number of planning iterations). Tuning over these hyperparameters (for both dense and sparse rewards) could lead to better performance, if desired.

For the first planning iteration we set $\Delta_{thres} = 0.01$, and for the later planning iterations, $\Delta_{thres} = 0.05$. We found that having a lower threshold for the first iteration helps the agent to avoid getting stuck in poor trajectories (i.e. avoid only using the policy), alongside the stochastic decision rule. For the Ant environment, we set $\Delta_{thres} = 0.01$ and always require at least one planning iteration.

### C.1.1    Sensitivity to Thresholds

We run a rough grid search with wider values for $\sigma_{thres}$ and $\epsilon_{thres}$, and calculate average reward in the Hopper changing worlds environment. The average reward for each setting is shown in Table C.1 and learning curves are shown in Fig. C.1. AOP is somewhat more sensitive to the setting of $\epsilon_{thres}$ early on in training, as a higher value corresponds to less planning, but this effect quickly dissipates. As a result, while the choice of $\sigma_{thres}$ and $\epsilon_{thres}$ is fairly arbitrary, we do not believe that AOP is particularly sensitive to them, and use the same values for all of the dense reward environments.

Table C.1: Effect of varying threshold hyperparameters

| Standard Deviation $\sigma_{thres}$: | 4 | 8 (Default) | 14 |
|---|---|---|---|
| Average reward | $0.47 \pm 0.20$ | $0.42 \pm 0.05$ | $0.44 \pm 0.16$ |
| **Bellman Error** $\epsilon_{thres}$: | **10** | **25 (Default)** | **40** |
| Average reward | $0.47 \pm 0.17$ | $0.47 \pm 0.09$ | $0.43 \pm 0.24$ |

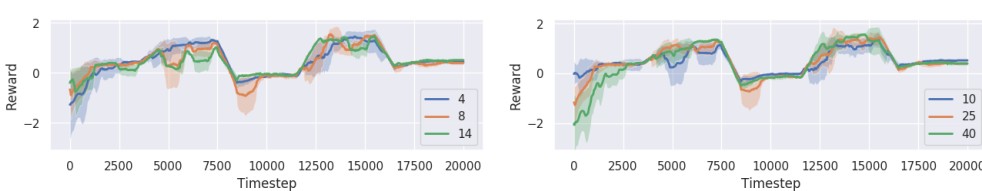

Figure C.1: Learning curves for hyperparameter sweep. Left: standard deviation $\sigma_{thres}$. Right: Bellman error $\epsilon_{thres}$. Legend shows value of relevant hyperparameter. 13 seeds were run in total.

## C.2 MODEL PREDICTIVE CONTROL HYPERPARAMETERS

Our MPPI temperature $\lambda$ is set to 0.01. The other planning hyperparameters (kept constant across environments) are shown below. See Section 3.4 for interpretation of policy optimization as a special case of AOP. Surprisingly, we found TD3 to perform worse with more than 1 trajectory per iteration.

| Parameter | AOP-TD3/AOP-BC | POLO | TD3 | PPO | MPC |
|---|---|---|---|---|---|
| Planning horizon | 1-80 | 80 | 256 | 128 | 80 |
| Planning iterations per timestep | 0-8 | 3 | 1 | 1 | 3, 8 |
| Trajectories per iteration | 40 | 40 | 1 | 32 | 40 |
| Noise standard deviation | 0.1 | 0.1 | 0.2 | - | 0.1 |

## C.3 NETWORK ARCHITECTURES

For our value ensembles, we use an ensemble size of 6 and $\kappa = 10^{-2}$. The value functions are updated in batches of size 32 for 32 gradient steps every 4 timesteps. All networks use tanh activations and a learning rate of $10^{-3}$, trained using Adam. Network sizes are shown below.

| Environment | AOP-TD3/AOP-BC | POLO | TD3 | PPO |
|---|---|---|---|---|
| Hopper | $V : (64, 64), Q : (400, 300), \pi : (400, 300)$ | $(64, 64)$ | $(400, 300)$ | $(64, 64)$ |
| Ant | $V : (64, 64), Q : (400, 300), \pi : (400, 300)$ | $(64, 64)$ | $(400, 300)$ | $(64, 64)$ |
| Maze | $(64, 64)$ | $(64, 64)$ | $(64, 64)$ | $(64, 64)$ |

## C.4 POLICY OPTIMIZATION HYPERPARAMETERS

Our TD3 uses the same hyperparameters as the original authors (Fujimoto et al., 2018b), where for every timestep, we run a rollout of length 256 and run 256 gradient steps. In the TD3 used for the experiment in Section 4.4, we run rollouts of length 1000 and run 1000 gradient steps after each rollout, equivalent to the standard TD3 setting with no termination.

Our PPO uses $\epsilon = 0.2$, $\lambda = 0.95$, batch sizes of 4096, and 80 gradient steps per iteration. For behavior cloning, we run 400 gradient steps on batches of size 64 every 4 timesteps. For the policy in AOP-TD3, we run 128 gradient steps on batches of size 100 every 4 timesteps.

## D ENVIRONMENT DETAILS

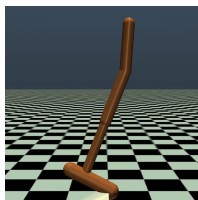 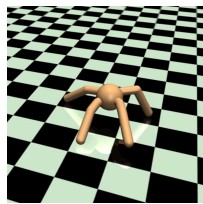 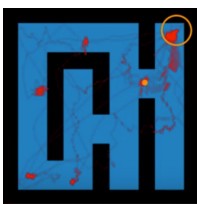

Figure D.1: Pictures of lifelong learning environments: Hopper, Ant, and Maze (from left to right). In the Maze, the agent (orange ball) must try to navigate to the goal (open orange circle) while avoiding the black walls. In the picture, the red lines indicate past history of movement.

In the online setting, the agent receives no signal from termination states, i.e. it becomes more difficult to know not to fall down in the cases of Hopper and Ant. To amend this, and achieve the same interpretable behavior as the standard reinforcement learning setting, we set the reward functions as the following for our environments, similar to Rajeswaran et al. (2017b):

| Environment | Reward Function |
|---|---|
| Hopper | $\lvert x\_vel - x\_vel_{targ} \rvert + 5(z - 1.8)^2 + .1 \lVert a \rVert_2^2 + x\_vel_{targ}$ |
| Ant | $\lvert x\_vel - 2 \rvert + 3(z - .9)^2 + .01 \lVert a \rVert_2^2$ |
| Maze (Dense) | $- \lVert (x, y) - (x, y)_{goal} \rVert_2 - \mathbf{1}\{\text{contact with wall}\}$ |
| Maze (Sparse) | $\mathbf{1}\{\text{inside goal}\} - \mathbf{1}\{\text{contact with wall}\}$ |

