# OpenReview forum: "Adaptive Online Planning for Continual Lifelong Learning"
_ICLR.cc/2020/Conference — Reject_

### Official Review · AnonReviewer1 · 2019-10-21
**Official Blind Review #1**

**Rating:** 1

**Review:**

The authors study continual, lifelong learning. They suggest a new algorithm, named Adaptive Online Planning (AOP) that combines model-based planning with model-free learning. AOP decides how much additional planning is needed based on the uncertainty of the model-free learner and the performance of the planner. Experiments are carried out on three tasks, i.e. Hopper, Ant and a Maze.

This paper should be rejected. The main reason is that the experiments were only performed for 3 different seeds and are therefore not statistically relevant (see Henderson et al. "Deep reinforcement learning that matters." Thirty-Second AAAI Conference on Artificial Intelligence. 2018.).

Besides the issue of significance of the results section, there are other concerns. Some of them are:
- Page 2: 'The dynamics model is updated immediately at world changes.' - Is this a reasonable assumption? Where does an accurate model come from? Given a perfect model, it is not surprising that a learner that is combined with such a model achieves a superior performance.
- Although the authors state that the 'ability to perform well in old tasks (backward transfer)' is important, they don't explicitly show their algorithm to achieve this goal. Backwards transfer might be included into the experimental section, but I could not find a statement that addresses this explicitly.
- I would like the authors to crisply define their use of the word 'online learning'. Does online learning simply mean to process each sample as it is available or does the term include real-time?
- How is \sigma_{thres} chosen? What is the influence of this parameter?
- The statement that 'AOP uses only 0 - 25% of the number of timesteps as MPC-8, but achieves generally comparable or stronger performance.' is wrong (see Fig 4, d and e). This statement is especially difficult, as results are only averaged over 3 runs.

There are furthermore a few minor concerns:
- the interval for \gamma should exclude 1 in this setting, as the return would otherwise be unbounded.
- In the background section, the authors confuse the definition of the return with reward.
- the term 'deep exploration' is used but not defined
- There are two figures between the subsection header for 4.4 and the text - this is highly confusing


**Experience Assessment:**

I have published in this field for several years.

**Review Assessment: Checking Correctness Of Derivations And Theory:**

I assessed the sensibility of the derivations and theory.

**Review Assessment: Checking Correctness Of Experiments:**

I carefully checked the experiments.

**Review Assessment: Thoroughness In Paper Reading:**

I read the paper thoroughly.

---

> ### Author Response · Authors · 2019-11-11
> **Response to Reviewer #1**
>
> Thank you for taking the time to read our paper and for providing feedback! We have added some details to the paper and hope to address some of your concerns:
>
> 1) Significance of results/seeds:
> We have updated all experiments to now include five seeds, as done in Henderson et al. 2018. In general, it is difficult for us to include more seeds due to computational constraints, but we hope this is satisfactory. In general, most of the AOP experiments are low-variance, as the standard deviations presented suggest.
>
> 2) Dynamics model:
> It is true that an accurate model is not a given in real-world robotics settings. However, we think the problem is still interesting. First, we would like to clarify that we compare AOP to other algorithms that also have access to an updated and correct dynamics model -- none of the algorithms discussed lack this access. Second, we believe that there are still many unsolved challenges and interesting ideas to consider, even with a perfect model. Control is still difficult, and learning control in a way that does not impact the agent’s future ability to learn is highly nontrivial. We observe that all algorithms struggle with this, even MPC, and notably PPO. If we cannot first do well in this setting with access to a model, then it would be extremely difficult to do so without one. Additionally, the idea of an agent that is not only knowledgeable about how to act, but also of when to plan, is not something that has been previously explored. Finally, some of the insights from our setting extend to other settings that are not obviously directly related: for example, in multi-agent settings, policies must be learned in a continually nonstationary environment, which we observe can be difficult with traditional methods, but can possibly be improved by strong exploration techniques -- multi-agent RL in some settings is deeply concerned with control, and not as much with learning the dynamics (in some settings they may even give agents access to other agents). We have further clarified some of the insights towards policy learning in a new Section 4.4.
>
> 3) Backwards transfer:
> We have now added an experiment demonstrating backwards transfer in the changing worlds Hopper environment: we take a trained policy from the end of AOP, and then train it in standard TD3 fashion in the initial setting (which it has not seen since the initial time). We compare this to a new policy (what is was when it first saw the world), and show that it adapts much more quickly, demonstrating backward transfer. Furthermore, we add some more analysis on the policies in general in Section 4.4.
>
> 4) Online learning:
> We define online learning as: for a particular timestep, the agent first trains on its own, and then is forced to make an action, before repeating for the next timestep. We have now further clarified this in our background section.
>
> 5) Threshold parameters:
> It is reasonable that any choice for \sigma_{thres} and \epsilon_{thres} is somewhat arbitrary; for our experiments, we picked a reasonable value around levels that the ensemble typically takes on throughout training. We have added an experiment to Appendix C.1.1 consisting of a grid search over a reasonable range of choices for these hyperparameters, and show that AOP is overall not particularly sensitive to them, so it is not especially important what we pick. In general, these parameters correspond to the algorithm’s inclination to cut planning.
>
> 6) Clarification on comparable performance/Ant environment:
> This statement refers to that, across most environments, AOP generally performs well, most of the time. We have amended this statement to clarify this. We would also like to discuss the Ant environment results in particular (old Figure 4 d & e): the Ant environment is particularly difficult, as most of the time the agent never gets up/takes a long time to get up after falling over, which showcases the sharp challenge of exploration in continual lifelong learning. Safe exploration is a well-studied topic, which we do not directly tackle, but is certainly an interesting problem to consider for future work in this setting. We have added more discussion on this in the “Vast Worlds” commentary in Section 4.3. *We do believe it is possible to improve this performance, and will likely post an update on it later this week.
>
> 7) Minor concerns:
> These have been corrected; in particular, we changed “deep exploration” to “temporally extended exploration”.
>
> Again, thank you for your feedback! Please let us know if you have other concerns, or topics you would like us to address/clarify further.

---

### Official Review · AnonReviewer2 · 2019-10-22
**Official Blind Review #2**

**Rating:** 6

**Review:**

The paper presents an adaptive online planning(AOP) strategy in a model-free policy setting, a reinforcement learning method aimed to solve catastrophic forgetting problem by combining model-based planning and model-free policy learning. AOP is able extensive plan only when necessary, leading to over all average reduced computation times. AOP can be easily integrated into other reinforcement learning frameworks such as to any offline-planning reinforcement learning algorithms.  The experiments demonstrate that AOP is computationally efficient compared to traditional baselines MPC-8 and MPC-3 while maintaining the performances.

The algorithm is developed based on heuristic solutions to address some of the fundamental problems in reinforcement learning, and although the proposed strategies definitely seem to provide some benefits in terms of computation complexity, the solution is not very elagant or noval. It is hard to justify the computational efficiency and performance in dynamically changing environments just based on the presented results. While the improvement in computation is there, what I find lacking is the experiments demonstrating clear evidence of overcoming catastrophic forgetting problem. The paper gives off a feeling that AOP as an add-on that can increase the performance of any  RL algorithm.


**Experience Assessment:**

I do not know much about this area.

**Review Assessment: Checking Correctness Of Derivations And Theory:**

N/A

**Review Assessment: Checking Correctness Of Experiments:**

I assessed the sensibility of the experiments.

**Review Assessment: Thoroughness In Paper Reading:**

I read the paper at least twice and used my best judgement in assessing the paper.

---

> ### Author Response · Authors · 2019-11-11
> **Response to Reviewer #2**
>
> Thank you for taking the time to read our paper and for providing feedback! We have added some details to the paper and hope to address some of your concerns:
>
> 1) Novelty of work:
> Our setting of continual lifelong learning has not been studied in the past, and is a new setup for which we analyze a new algorithm and adaptations of existing algorithms on. Additionally, past work into reducing the computation of a planner has been limited. We kindly ask that you consider our work in the broader context of our setting.
>
> 2) Catastrophic forgetting:
> We have now added an experiment demonstrating backwards transfer in the changing worlds Hopper environment: we take a trained policy from the end of AOP, and then train it in standard TD3 fashion in the initial setting (which it has not seen since the initial time). We compare this to a new policy (what is was when it first saw the world), and show that it adapts much more quickly, demonstrating backward transfer. Furthermore, we add some more analysis on the policies in general in Section 4.4.
>
> 3) Increase in performance for any RL algorithm:
> There are many algorithms that can be fit into the AOP framework; however, we think that it is important to note that the goal of AOP is not directly to increase performance, but rather primarily to reduce computation, and in some cases improve exploration.
>
> Again, thank you for your feedback! Please let us know if you have other concerns, or topics you would like us to address/clarify further.

---

### Official Review · AnonReviewer3 · 2019-10-22
**Official Blind Review #3**

**Rating:** 3

**Review:**


The work is heuristically motivated by the goal of reducing the high computation of model-based learning while achieving high performance. For achieving that, the authors propose an algorithm, Adaptive Online Planning (AOP) combining a model-free policy learning method and a model-based planner. In terms of the empirical study, they test the algorithm in 3 environments, Hopper, Ant, and Maze. They compare their algorithms with several model-based methods.

From my perspective, the paper has several weaknesses for which I give a weak rejection.

The motivation is interesting to me, but the authors do not provide enough justification. The authors claim that the proposed method is able to reduce high computation. However, seemingly they only intuitively illustrate how it saves energy without strong proofs, which weakens the claim. What’s more, the experiment is not clear to me. What are the take-aways of Figure 3 and Figure 4 while I cannot see an improvement from them? There is no comparison in Figure 6; not clear how the plots of other models look like. The last comment is about the 3 environments that are not complex enough.


Minor comments:
- Some typos and grammar mistakes, e.g., ‘planing’ and ‘(d)by’ in the third last line (p.4); the second sentence in Sec. conclusion (p.8).

**Experience Assessment:**

I do not know much about this area.

**Review Assessment: Checking Correctness Of Derivations And Theory:**

I did not assess the derivations or theory.

**Review Assessment: Checking Correctness Of Experiments:**

I assessed the sensibility of the experiments.

**Review Assessment: Thoroughness In Paper Reading:**

I read the paper at least twice and used my best judgement in assessing the paper.

---

> ### Author Response · Authors · 2019-11-11
> **Response to Reviewer #3**
>
> Thank you for taking the time to read our paper and for providing feedback! We have added some details to the paper and hope to address some of your concerns:
>
> 1) Theoretical justification of algorithm and motivation:
> While we do not provide theoretical justification for AOP, our main contribution is the introduction of a new problem setting, and the proposal of an initial idea to tackle it. This problem has close ties to nonstationary environments, which are broadly relevant in many settings, ex. multi-agent settings, policy learning in learned dynamics, real-world robotics where resets are costly, etc. Furthermore, we identify several challenges in such a setting, and show where previous methods fail, which can lead to insights on how to improve methods more generally. We hope you will consider our contribution as whole.
>
> 2) Takeaways from Figures 3/4 [now Figures A.1 and A.2] (computation/rewards):
> We agree that the graphs are difficult to see information from; we have now summarized the information compactly, moved the detailed graphs into Appendix A, and added some clarifications on takeaways from the experiments. For the particularly interesting takeaway of policy degradation, we have kept the relevant graphs and added further discussion in Section 4.4. We hope this is now more clearly showing the reduction in computation and the strength of performance of the model-based/model-free algorithms.
>
> 3) Comparisons in Figure 6 [now Figure 5] (behavior of AOP):
> We dedicate Section 4.5 to discussing the specific components of the AOP algorithm, namely individual statistics (Bellman error and standard deviation of the value ensemble, planning horizon length, planning iterations, policy usage) that help to give a clearer picture of what the algorithm is doing at each stage of training. Therefore, we do not plot other algorithms on the same graph. Notably, uncertainty and planning decrease as the agent progresses farther in each world.
>
> 4) Complexity of environments:
> It is true that the environments themselves are not particularly complex control environments, and have been solved adequately in the past in the offline setting. However, we show that these environments become problematic for state-of-the-art algorithms (TD3, PPO, POLO) when tackled in continual lifelong learning, due to the lack of ability to reset, nonstationary dynamics, etc. Therefore, we believe that they are complex enough for our investigations, and are capable of crisply showing where existing work struggles.
>
> Again, thank you for your feedback! Please let us know if you have other concerns, or topics you would like us to address/clarify further.

---

### Author Response · Authors · 2019-11-11
**Summary of Changes**

We would like to thank all of the reviewers for their responses; we have left specific comments in individual responses. We summarize here changes made in our updated version of the paper (11/11/19):

- All experiments are now run with 5+ seeds (from old number of 3 seeds)
- Changed main experimental results in Section 4 to be presented in table form, and moved the per-timestep graphs into Appendix A
- Added new experiments in Section 4.4, highlighting policy degradation and backwards transfer effects in a simpler, standard episodic context
- Added a new hyperparameter grid search in Appendix C.1.1, showing the robustness of AOP to choices of thresholds
- Further discussion of results in Section 4.3, Challenges in Continual Lifelong Learning Setting: notably the difficulty of Ant and the additional learning of AOP in sparse maze
- Restructured appendix, added some new details
- Minor typos fixed, small wording changes in various sections

We hope that these address most of the concerns. From a big picture, our work is broadly a study into a new continual lifelong learning setting, and additionally the proposal of an algorithm that performs well in this setting -- we would kindly like to ask that our paper be evaluated in this context. Please let us know if there are any remaining concerns or topics that you would like us to address.

(We are planning to release an additional update before the end of the review period).

---

### Author Response · Authors · 2019-11-15
**Final Summary of Changes**

Our replies to specific concerns were left in the comments below. We show a summary of our total changes since the paper was first submitted (> indicates change after 11/11 (after last summary), - indicates change before 11/11 (in last summary)):

> There was an issue with the Ant environment causing learning to be unstable for all algorithms, which is now fixed, and experiments were updated
- All experiments are now run with 5+ seeds (from old number of 3 seeds)
- Added new experiments in Section 4.4, highlighting policy degradation and backwards transfer effects in a simple episodic context
- Added a hyperparameter grid search in Appendix C.1.1, showing robustness of AOP to choice of thresholds
- Further discussion of results in Section 4.3, Challenges in Continual Lifelong Learning Setting to clarify results/takeaways from experiments
- Moved main experimental graphs to Appendix A, and instead summarized them compactly in Tables 1 & 2
- Minor typos fixed, wording changes in various sections

To summarize some of our past responses: we introduced a novel reinforcement learning setting closer to real world usage, showed that existing approaches can fail even with access to a ground truth dynamics model, and proposed a new algorithm for success in this setting. We only utilize the dynamics model locally, which represents strong learning of a model around recent data collected by a policy; even when we assume this model is perfect, TD3 and PPO still fail. Our algorithm uses around one-tenth of the planning of MPC, and a third of POLO -- both strong ground truth baselines -- and achieves comparable performance in most settings. The environments, though not complex in the standard offline RL setting, become extremely difficult in our continual lifelong learning setting.

We thank the reviewers and area chair for the time spent reviewing our work, and would appreciate if the reviews could be updated if our responses have been satisfactory.

---

### Decision · Program_Chairs · 2019-12-19

**Decision:**

Reject

**Comment:**

A new setting for lifelong learning is analyzed and a new method, AOP, is introduced, which combines a model-free with a model-based  approach to deal with this setting.

While the idea is interesting, the main claims are insufficiently demonstrated. A theoretical justification is missing, and the experiments alone are not rigorous enough to draw strong conclusions. The three environments are rather simplistic and there are concerns about the statistical significance, for at least some of the experiments.